Genome sequencing and analysis of the first complete genome of Lactobacillus kunkeei strain MP2, an Apis mellifera gut isolate

Asenjo Freddy 1
Olmos Alejandro 2
Henríquez-Piskulich Patricia 3
Polanco Victor 2 3
Aldea Patricia 3
Ugalde Juan A. 1 jugalde@udd.cl
Trombert Annette N. 2 annette.trombert@umayor.cl
1 Centro de Genética y Genómica, Facultad de Medicina, Clinica Alemana Universidad del Desarrollo , Santiago , Chile
2 Centro de Genómica y Bioinformática, Facultad de Ciencias, Universidad Mayor , Santiago , Chile
3 Centro de Estudios Apícolas CEAPI Mayor, Facultad de Ciencias, Universidad Mayor , Santiago , Chile
Landa Blanca
Electronic publication date: 2016 Apr 19
Publication date: 2016
Volume: 4
Electronic Location ID: e1950
Received 2015 Oct 19; Accepted 2016 Mar 29
Copyright: ©2016 Asenjo et al.
Copyright year: 2016
Copyright holder: Asenjo et al.
License: This is an open access article distributed under the terms of the Creative Commons Attribution License, which permits unrestricted use, distribution, reproduction and adaptation in any medium and for any purpose provided that it is properly attributed. For attribution, the original author(s), title, publication source (PeerJ) and either DOI or URL of the article must be cited.
License URL: https://creativecommons.org/licenses/by/4.0/

Keywords: Lactobacillus, Comparative genomics, Apis mellifera, Honeybee

Funding: Bees for Care B4C 13CTI-21546 Conicyt Grant Fondeyct Iniciación No 11140666 This work was supported by the project Bees for Care (B4C 13CTI-21546) and a Conicyt grant (Fondeyct Iniciación No 11140666). The funders had no role in study design, data collection and analysis, decision to publish, or preparation of the manuscript.

==============================
Background. The honey bee (Apis mellifera) is the most important pollinator in agriculture worldwide. However, the number of honey bees has fallen significantly since 2006, becoming a huge ecological problem nowadays. The principal cause is CCD, or Colony Collapse Disorder, characterized by the seemingly spontaneous abandonment of hives by their workers. One of the characteristics of CCD in honey bees is the alteration of the bacterial communities in their gastrointestinal tract, mainly due to the decrease of Firmicutes populations, such as the Lactobacilli. At this time, the causes of these alterations remain unknown. We recently isolated a strain of Lactobacillus kunkeei (L. kunkeei strain MP2) from the gut of Chilean honey bees. L. kunkeei, is one of the most commonly isolated bacterium from the honey bee gut and is highly versatile in different ecological niches. In this study, we aimed to elucidate in detail, the L. kunkeei genetic background and perform a comparative genome analysis with other Lactobacillus species.

Methods. L. kunkeei MP2 was originally isolated from the guts of Chilean A. mellifera individuals. Genome sequencing was done using Pacific Biosciences single-molecule real-time sequencing technology. De novo assembly was performed using Celera assembler. The genome was annotated using Prokka, and functional information was added using the EggNOG 3.1 database. In addition, genomic islands were predicted using IslandViewer, and pro-phage sequences using PHAST. Comparisons between L. kunkeei MP2 with other L. kunkeei, and Lactobacillus strains were done using Roary.

Results. The complete genome of L. kunkeei MP2 comprises one circular chromosome of 1,614,522 nt. with a GC content of 36,9%. Pangenome analysis with 16 L. kunkeei strains, identified 113 unique genes, most of them related to phage insertions. A large and unique region of L. kunkeei MP2 genome contains several genes that encode for phage structural protein and replication components. Comparative analysis of MP2 with other Lactobacillus species, identified several unique genes of L. kunkeei MP2 related with metabolism, biofilm generation, survival under stress conditions, and mobile genetic elements (MGEs).

Discussion. The presence of multiple mobile genetic elements, including phage sequences, suggest a high degree of genetic variability in L. kunkeei. Its versatility and ability to survive in different ecological niches (bee guts, flowers, fruits among others) could be given by its genetic capacity to change and adapt to different environments. L. kunkeei could be a new source of Lactobacillus with beneficial properties. Indeed, L. kunkeei MP2 could play an important role in honey bee nutrition through the synthesis of components as isoprenoids.

Introduction

The honey bee (Apis mellifera) is the most important pollinator in agriculture worldwide, playing a key role in the human food supply by providing pollination services for diverse crops (Evans & Schwarz, 2011). However, from 2006 to this day, an unusual decrease in honey bee colonies has been taking place, known as Colony Collapse Disorder (CCD). CCD describes the seemingly spontaneous abandonment of the hives by honey bee workers, where queens often stay in the hive accompanied by a small group of nurse worker bees. The specific causes of CCD are unknown, but several factors can impact the health of honey bees, and contribute to this phenomenon: (1) pests and diseases (such as, American foulbrood, European foulbrood, chalkbrood nosema, small hive beetles, and tracheal mites); (2) the use of chemicals in bee colonies, and their surrounding environment; (3) beekeeping practices; (4) agricultural practices and (5) climate change (Henry et al., 2012; Di Pasquale et al., 2013; Di Prisco et al., 2013).

Multiple studies have suggested that CCD directly affects the microbial composition of the honey bee gut microbiota. Eight dominant groups can be found in the honey bee gut (Cox-Foster et al., 2007; Martinson et al., 2011): Gammaproteobacteria (Enterobacteriaceae and Pasteurellaceae), Betaproteobacteria (Neisseriaceae), Alphaproteobacteria (Rhizobiales, Acetobacteraceae), Firmicutes (Lactobacillus sp.), and Actinobacteria (Bifidobacterium sp.) groups (Cox-Foster et al., 2007; Martinson et al., 2011). Gut microbiome studies from individuals obtained from colonies affected and non-affected by CCD, indicated an increase in the Gammaproteobacteria, and a decrease of the Firmicutes in affected colonies, showing how the CCD condition affects commensal communities in the honey bee gut (Cox-Foster et al., 2007). Firmicutes includes Gram-positive and low-G + C bacteria, such as the Lactobacillus genus, where some of its members have been implicated in the fitness improvement of honey bees (Audisio & Benítez-Ahrendts, 2011; Audisio, Sabaté & Benítez-Ahrendts, 2015).

The study of lactobacilli members of the honey bee microbiota can give us information about beneficial species for honey bees. One of the most common lactobacilli species present in the honey bee gut microbiota is Lactobacillus kunkeei, described for the first time as a spoilage organism isolated from commercial grape wine (Edwards et al., 1998). Characterized as a Gram-positive fructophilic lactic acid bacterium (FLAB), L. kunkeei possesses a weak catalase activity and has the ability to ferment carbohydrates, such as glucose, fructose, sucrose, raffinose and mannitol but prefer fructose. The fermentation products of these reactions are lactic acid and acetic acid (Edwards et al., 1998; Bae, Fleet & Heard, 2006; Endo, Futagawa-Endo & Dicks, 2009; Endo, 2012). L. kunkeei can be found in fructose rich-niches, including honey, beebread, wine and flowers (Vásquez et al., 2012; Endo et al., 2012). Furthermore it is present in the gastrointestinal tract of several insects found in flowers, such as tropical fruit flies, Camponotus spp (carpenter ants), bumblebees and honey bees (Neveling, Endo & Dicks, 2012; Anderson et al., 2013; Endo & Salminen, 2013). During the summer months, L. kunkeei is the most frequent lactobacilli isolate from the honey bee gut (Corby-Harris, Maes & Anderson, 2014; McFrederick et al., 2014).

In previous work, we isolated a L. kunkeei strain (named as MP2) from the gut of Chilean honey bees from María Pinto, Melipilla (Olmos et al., 2014). The genome of this strain was sequenced using the Illumina MiSeq platform, which resulted in a draft genome of 44 contigs, for a total genome size of 1,581,395 bp, and 826 well-annotated protein coding-genes (Olmos et al., 2014). The nature of the short-reads used for this assembly did not allow for complete resolution of the genome without gaps. In addition, multiple repetitions, including the presence of multiple copies of the ribosomal operon, could not be resolved in this draft genome. To overcome these limitations, we performed a re-sequencing of the L. kunkeei MP2 genome, using single molecule sequencing in the Pacific Biosciences platform.

In this work, we report the first complete genome sequence of L. kunkeei MP2, its characterization, and comparison with other Lactobacillus genomes.

Methods

DNA isolation and genome sequencing

The L. kunkeei strain MP2 reported in this study was isolated in a previous study from the gut of a honey bee (Apis mellifera), collected from commune hives in the Maria Pinto area, Melipilla Province in the Central zone of Chile (Olmos et al., 2014). Collection and use of honey bees for these studies was reviewed and approved by the Bioethics Committee of Universidad Mayor, which is governed by the regulations of the Animal Health Services of Chile.

For DNA extraction, L. kunkeei MP2 colonies were cultured in MRS broth (37 °C, 5% CO2) and genomic DNA was obtained using a silica-based protocol (Boom et al., 1999). Briefly, bacterial pellet was lysed using a solution composed of SDS 10%, proteinase K (10 mg/mL, Thermo Scientific) and lysozyme (5 mg/mL, Pierce) at 37 °C for 60 min. The lysate was mixed with guanidine chloride 6M and a silica suspension (50% w/v) and incubated for 10 min. The silica was centrifuged, and DNA was recovered, after washes with 70% alcohol, into sterile, free nuclease water. Approximately 13.5 µg of DNA were used to construct sequencing libraries with an average insert size of 20 kb, and sequenced using one SMRT cell (P6-C4 Chemistry) on a PacBio RSII sequencer (Pacific Biosciences) at the UCSD IGM Genomics Center.

Genome assembly and annotation

Raw reads (∼1 Gbps) were processed to remove SMRT bell adapters, short and low-quality reads (<80% accuracy) using SMRT Analysis version 2.3. A total of 154,044 filtered reads (average length, 9 Kb) were used for de novo assembly using Celera Assembler version 8.3 (Myers et al., 2000), with self-correction of the PacBio reads (Berlin et al., 2015). Polishing was done using Quiver, using SMRT Analysis version 2.3. Comparisons between the previously sequenced L. kunkeei MP2 genome (Accession number PRJNA257367) (Olmos et al., 2014), as well with the other two available genome sequences (Porcellato et al., 2015; Djukic et al., 2015) were performed using MUMMER (Kurtz et al., 2004). Genome annotation was performed using Prokka version 1.11 (Seemann, 2014). The predicted CDS were classified into EggNOG categories using HMMER version 3.1 (http://hmmer.org) against the EggNOG 4.1 database (Powell et al., 2014) with an E-value cutoff of 1E-05. Genomic islands were annotated using IslandViewer 3 (Dhillon et al., 2015), and possible phage sequences were searched using PHAST (Zhou et al., 2011). Genome visualization was done using Circos version 0.69 (Krzywinski et al., 2009). The genome sequence an assembly is available at NCBI with the accession number PRJNA298292.

Pan-genome analysis

Comparative genomic analysis was performed from a list of selected genomes from Lactobacillus species, as well as other strains of L. kunkeei (Table 1). To avoid possible biases in the comparisons due to different annotation procedures, all of the genomes were re-annotated using Prokka version 1.11 (Seemann, 2014). Comparisons between all the selected genomes, were done using Roary (Page et al., 2015), with a blast identity cutoff of 97% for the comparison between L. kunkeei strains, and a 40% for the comparison between Lactobacillus species. In addition at the genus level, paralog splitting was disabled. Visualization of the pan-genome data was done using Anvi’o (Eren et al., 2015).

Table 1 Genome sequences used in this study.

Species name and accession numbers of the genomes selected in this study.

Genomes	Bioproject	Assembly	
Lactobacillus kunkeei DSM 12361	PRJNA222257	GCA_001433825.1	
Lactobacillus kunkeei Fhon2	PRJNA270967	GCA_001281165.1	
Lactobacillus kunkeei LAan	PRJNA270961	GCA_001281225.1	
Lactobacillus kunkeei LAce	PRJNA270962	GCA_001421115.1	
Lactobacillus kunkeei LAfl	PRJNA270964	GCA_001421135.1	
Lactobacillus kunkeei LAko	PRJNA270965	GCA_001281205.1	
Lactobacillus kunkeei LAla	PRJNA270966	GCA_001281215.1	
Lactobacillus kunkeei LAni	PRJNA270969	GCA_001281285.1	
Lactobacillus kunkeei LMbe	PRJNA270972	GCA_001308185.1	
Lactobacillus kunkeei LMbo	PRJNA270973	GCA_001308195.1	
Lactobacillus kunkeei LAdo	PRJNA270963	GCA_001308205.1	
Lactobacillus kunkeei LAnu	PRJNA270970	GCA_001308215.1	
Lactobacillus kunkeei EFB6	PRJNA227106	GCA_000687335.1	
Lactobacillus kunkeei AR114	PRJNA253911	GCA_000830375.1	
Lactobacillus kunkeei YH-15	PRJNA270974	GCA_001281265.1	
Lactobacillus kunkeei MP2	PRJNA298292	GCA_001314945.1	
Lactobacillus acidophilus 30SC	PRJNA63605	GCA_000191545.1	
Lactobacillus amylovorus GRL1118	PRJNA160233	GCA_000194115.1	
Lactobacillus brevis KB290	PRJNA195560	GCA_000359625.1	
Lactobacillus buchneri NRRL B-30929	PRJNA66205	GCA_000211375.1	
Lactobacillus casei BD-II	PRJNA162119	GCA_000194765.1	
Lactobacillus delbrueckii subsp. bulgaricus ND02	PRJNA60621	GCA_000182835.1	
Lactobacillus fermentum CECT 5716	PRJNA162003	GCA_000210515.1	
Lactobacillus gasseri 130918	PRJNA224116	GCA_000814885.1	
Lactobacillus helveticus H10	PRJNA162017	GCA_000189515.1	
Lactobacillus hokkaidonensis JCM 18461	PRJNA224116	GCA_000829395.1	
Lactobacillus johnsonii DPC 6026	PRJNA162057	GCA_000204985.1	
Lactobacillus kefiranofaciens ZW3	PRJNA67985	GCA_000214785.1	
Lactobacillus mucosae LM1	PRJNA86029	GCA_000248095.3	
Lactobacillus paracasei subsp. paracasei 8700:2	PRJNA55295	GCA_000155515.2	
Lactobacillus plantarum	PRJNA224116	GCA_000931425.1	
Lactobacillus reuteri SD2112	PRJNA55357	GCA_000159455.2	
Lactobacillus rhamnosus LOCK908	PRJNA210958	GCA_000418495.1	
Lactobacillus ruminis ATCC 27782	PRJNA73417	GCA_000224985.1	
Lactobacillus sakei subsp. sakei 23K	PRJNA58281	GCA_000026065.1	
Lactobacillus salivarius	PRJNA224116	GCA_000758365.1	
Lactobacillus sanfranciscensis TMW 1.1304	PRJNA72937	GCA_000225325.1	
Lactobacillus sp. wkB8	PRJNA224116	GCA_000761135.1	

Phylogenetic analysis

16S rRNA gene sequences were obtained from the Silva database (Quast et al., 2013), and aligned using the SINA webserver aligner (Pruesse, Peplies & Glöckner, 2012) with default parameters, and using the Bacteria variability profile. For the L. kunkeei strain tree, we used the core genome (genes shared by all L. kunkeei strains), aligning the genes using Prank (Löytynoja, 2014). For both cases, the phylogenetic trees were generated using FastTree v 2.1.7 (Price, Dehal & Arkin, 2010), with the–slow option.

Results and Discussion

Assembly description

The PacBio reads obtained for L. kunkeei MP2 were assembled using MHAP (Berlin et al., 2015) implemented in the PBcR pipeline (Celera Assembler 8.3) (Myers et al., 2000). This de novo assembly resulted in one contig, representing the complete genome of L. kunkeei MP2 in a single 1,614,522 nt chromosome. A total of 1,468 CDS were predicted in the genome, 67 tRNA and 5 copies of the ribosomal operon. Functional annotation was done using EggNOG V 4.1 (Powell et al., 2014), and the summary of functional categories is shown on Table 2 (gene annotation on Table S1). The %GC content of the genome was 36.9%, and several features of interests, such as the presence of prophage regions, were found. With this assembly, we were able to differentiate the three ribosomal operons that are present in the chromosome, something that was not possible in the previous sequenced genome of this strain (Olmos et al., 2014).

Table 2 EggNOG functional categories for the predicted genes of L. kunkeei MP2.

Information Storage and Processing		
Translation, ribosomal structure and biogenesis	127	
Transcription	73	
Replication, recombination and repair	129	
Cellular Processes and Signaling		
Cell cycle control, cell division, chromosome partitioning	22	
Defense mechanisms	15	
Signal transduction mechanisms	23	
Cell wall/membrane/envelope biogenesis	77	
Cell motility	4	
Intracellular trafficking, secretion, and vesicular transport	19	
Posttranslational modification, protein turnover, chaperones	45	
Metabolism		
Energy production and conversion	41	
Carbohydrate transport and metabolism	58	
Amino acid transport and metabolism	107	
Nucleotide transport and metabolism	69	
Coenzyme transport and metabolism	25	
Lipid transport and metabolism	32	
Inorganic ion transport and metabolism	65	
Secondary metabolites biosynthesis, transport and catabolism	8	
Poorly Characterized		
Function unknown	414	

A comparison of the assembly of L. kunkeei MP2 obtained in this work, with the previously obtained using Illumina sequencing (Olmos et al., 2014), is shown in Fig. 1. All of the previous assembled contigs mapped to the current assembly, and several gaps on the sequence were completed in this new version of the genome.

Figure 1 Genome organisation of L. kunkeei MP2.

Circular overview of the complete genome of L. kunkeei MP2, highlighting some of the features. Starting from the outside ring towards the interior: EggNOG annotation of the predicted CDS; Contig recruitment of the previous L. kunkeei MP2 genome sequencing (Olmos et al., 2014); Phage island predictions using Island Viewer 3; Unique genes of L. kunkeei MP2, compared to 16 strains of L. kunkeei; Unique genes of L. kunkeei MP2 compared with 22 genomes of Lactobacillus species; %GC contento of the L. kunkeei MP2 genome.

Central Metabolism of L. kunkeei MP2

Energy metabolism

MP2 has the complete route for acetate synthesis, with the presence of the gene codifying for phosphoglycerate kinase. No genes codifying for phosphoribulokinase (PRK) and ribulose-biphosphate carboxylase (RbcL), two of the enzymes involved in the synthesis of glyceraldehyde-3-phosphate synthesis, were found on the genome.

Carbohydrate metabolism

The genes that encode for the enzymes phosphofructokinase/glucokinase (PFK) and Fructose-biphosphate aldolase (FBA), were not found in the genome of L. kunkeei MP2. These enzymes are part of the Embden-Meyerhof pathway and are involved in the homofermentative metabolism of lactic acid. As a fructophilic bacterium, L. kunkeei MP2 can synthesize ribose-5-phosphate through pentose phosphate pathway from fructose and obtain PRPP (phosphoribose pyrophosphate), the precursor of purine, pyrimidine and histidine metabolism. For the synthesis of ribose-5-phosphate, L. kunkeei uses the route from B-D-fructose-6-phosphate through D-arabino-Hex-3-ulose-6-phosphate intermediate. L. kunkeei MP2 can synthesize UDP-glucose and has two isoprenoid biosynthesis pathways, the mevalonate and the non-mevalonate pathways. Isoprenoids include carotenoids, sterols, prenyl side-chains of chlorophylls, and plastoquinone, exhibiting many biological functions (Daum et al., 2009). In whiteflies (Bemisia tabaci), the genome of its endosymbiotic bacteria, Candidatus Portiera aleyrodidarum, encodes for key enzymes in carotenoids synthesis, suggesting that whitefly not only can acquire carotenoids from the diet, but also from their microbiota (Sloan & Moran, 2012). Therefore, if L. kunkeei produces key enzymes involved in isoprenoid synthesis, it is possible that it could be playing an important role in honey bee nutrition.

Nucleotide and amino acid metabolism

The pathways for purine biosynthesis are complete. However, in the de novo pirimidine pathway, L. kunkeei lacks the gene pyrB, which codifies for the aspartate carbamoyl transferase, and ndk, codifying for the nucleoside diphosphate kinase. The analysis of the metabolic pathways in MP2 revealed a minimal amino acid auxotrophy (methionine or cysteine), with the presence of the genes that encode for a D-methionine transport system, suggesting the ability of L. kunkeei MP2 to acquire methionine/cysteine from the environment. These results are in line with previous reports of the lactobacilli being auxotrophic for both methionine and cysteine (Seefeldt & Weimer, 2000), and where the supplement of culture media with these amino acids improved bacterial growth (Lozo et al., 2008). A gene that encodes for serine hydroxymethyltransferase (SHMT) was found in the genome of MP2. This enzyme catalyzes the addition of formaldehyde to glycine, a key step for the production of serine (Jiang et al., 2014), and appears to be absent in the other Lactobacillus genomes analyzed in this study. Its presence in L. kunkeei MP2 could be part of specific adaptation mechanisms of this species to its environment.

Prophage insertions

Previous work in other Lactobacillus species, reported the presence of regions with prophage genes in their genomes, including species such as L. rhamnosus, L. gasseri, L. salivarius, L. casei, L. lactis, and L. johnsonii (Ventura et al., 2004; Ventura et al., 2006; Kankainen et al., 2009; Savabi et al., 2014; Baugher, Durmaz & Klaenhammer, 2014). This shows the widespread abundance of prophages in the genomes of Lactobacillus species, a characteristic shared by L. kunkeei MP2. Two regions were identified by PHAST (Zhou et al., 2011), as putative prophage insertions (Table S2). One of them, located in the region between 594,506 and 613,136, was found to be present in all the 23 Lactobacillus genomes used in this work. The second region, located around 32,973–75,092, was found to be unique to L. kunkeei MP2, compared to other strains of L. kunkeei, as well as other Lactobacillus species. In at least one Lactobacillus species (L. gasseri), the presence of these inserted phages has been associated with the horizontal transfer of genes (Baugher, Durmaz & Klaenhammer, 2014), suggesting a possible role for these elements within the genome of L. kunkeei MP2. However, the detailed mechanisms, as well as the possible adaptive consequences of such events, need to be explored in more detail in the future.

Comparison of L. kunkeei MP2 with other L. kunkeei strains

We performed a comparative genomic analysis of MP2 against sixteen publicly available genome sequences of L. kunkeei strains (Table 1). This analysis can provide us with a snapshot of the unique features that are present in this strain, based on its gene content. Recent work, highlighted the important role that genome reduction played in the evolution of L. kunkeei (Tamarit et al., 2015), which suggests that we would expect to find only a few genes truly unique to L. kunkeei MP2, compared to other strains.

Pangenome analysis resulted in the identification of a set of 853 core genes (present in all the strains). For the shell genes, we found that 813 genes are found between 15% to 95% of the analyzed strains, while 1661 genes are present in less than 15% of the strains. Focusing on the MP2 strain, based on this analysis we identified 113 genes that are not present in any of the other strains, representing close to 7.7% of the total number of genes.

Using the pangenome matrix, we can visualize the relationships between the genomes based only on the clustering of the proteins (presence or absence of a gene in a protein group). Visualizations of the results (Fig. 2), show that the strains LAni, LAce, LAan and EFB6, cluster in the same group as MP2, based on their gene content. This correlates with the phylogenetic placement of MP2 within the same group, based on a concatenated alignment of the core genome (853 genes; 807,585 nucleotides) (Fig. 3). The closest strain on the tree is EFB6, which was also isolated from the gut of A. mellifera (although from a larvae affected by European foulbrood) (Djukic et al., 2015). Currently there is no evidence of association between particular strains and a particular species of Apis (Tamarit et al., 2015). A better understanding of the interactions between the gut microbiota and the host needs to incorporate additional variables, such as the surrounding environment (including flowers and fruit), seasonal variations, among other possible elements. These factors could play a strong role in the association between an Apis species and a L. kunkeei strain.

Figure 2 Anvi’o pangenome visualization of 16 L. kunkeei genomes.

The outer core in red, shows the core genome of L. kunkeei, protein clusters shared among all the strains (853 genes). The genomes are clustered based on the presence/absence pattern of protein clusters. MP2 is highlighted in green, while the more similar strains based on the clustering pattern, are highlighted in lighter green (LAni, LAce, LAan, and EFB6). Pan-genome visualization was generated using Anvi’o (Eren et al., 2015).

Figure 3 Phylogenetic tree of 16 L. kunkeei strains.

The tree was constructed using all of the genes shared between all 16 strains (853 genes; 807,585 nucleotides).

One of the main differences of MP2 compared to the other L. kunkeei genomes, is the presence of multiple phage genes inserted in several parts of the genome. One of these unique phage regions can be found at coordinates 31,034–75,092 (Fig. 1). It is a large region, which includes several phage-related proteins, including structural and replications components. Sequence analysis using Blast, shows that these proteins are related to phages that infect Gram-positive Bacteria, such as Bacillus (Hastings et al., 2013), Listeria (Dorscht et al., 2009), Enterococcus (Yasmin et al., 2010), and Staphylococcus (Chang et al., 2013) (Table S3).

Comparison of L. kunkeei MP2 with other Lactobacillus strains

Phylogenetic reconstructions using the sequence that encodes for the 16S rRNA gene, shows that the closes species to L. kunkeei MP2 is L. sanfranciscensis (Fig. 4). This placement is in agreement with previous diversity analysis performed on acidophilic bacteria (McFrederick et al., 2012), relating the taxonomy of both species.

Figure 4 Phylogenetic tree of several Lactobacillus species, including L. kunkeei MP2.

Phylogenetic reconstruction was done using the sequence of the 16S rRNA gene.

Whole genome comparisons between L. kunkeei MP2 and other Lactobacillus species, discovered several unique genes. One example is gtfC, which encodes for a glucosyltransferase, which has been extensively studied in Streptococcus mutans, where is expressed in the presence of carbohydrates such as sucrose, D-glucose, D-fructose, among others (Shemesh et al., 2006). GtfC (as well as GftB), is also considered a virulence factor in S. mutants, promoting bacterial adhesion to smooth surfaces and cells (Tsumori & Kuramitsu, 1997). Also, GtfC is part of the synthesis route of a mixture of insoluble and soluble glucans, which are important components of cariogenic biofilms (Yousefi et al., 2012). Considering the rich carbohydrate environment where L. kunkeei can survive, the presence of unique glucosyltransferase genes, such as gftC, could facilitate bacterial colonization of flowers, as well as the honey bee gut.

Another unique gene found in L. kunkeei, encodes for the adapter protein MecA, a pleiotropic regulator of bacterial development. This protein has been shown to affect competence, protein degradation and sporulation in Bacteria, such as Bacillus subtilis (Schlothauer et al., 2003). MecA interacts with the chaperone ClpC, and with the transcription factor ComK, promoting the degradation of this protein during the logarithmic growth phase. The degradation of ComK stops when bacteria enters to stationary growth phase, where the quorum-sensing pheromone ComX promotes the synthesis of ComS, which binds to MecA and prevents the interaction of MecA-ComK (Persuh, Mandic-Mulec & Dubnau, 2002; Prepiak et al., 2011; Wahl et al., 2014). This could have an effect on the biofilm generation capabilities of L. kunkeei MP2, but this needs to be explored experimentally.

At least seven different Lactobacillus species have been characterized in the gut microbiota of A. mellifera, where it has been suggested that they play different roles in the stability of the host functions (Engel & Moran, 2013). L. kunkeei MP2 appears to have a unique set of genes when compared to other strains of L. kunkeei, as well as with other species of Lactobacillus (Table S3), which suggest unique adaptation strategies of L. kunkeei MP2 to the gut of A. mellifera.

We also identified a hypothetical protein with similarities to a low-molecular-weight protein-tyrosine phosphatase (LMPTP), unique to the L. kunkeei MP2 genome, compared to other L. kunkeei strains and other Lactobacillus species. This LMPTP is similar to the YfkJ protein from Bacillus subtilis, which has been involved in the response to ethanol stress (Musumeci et al., 2005). Ethanol, and other organic compounds, are commonly present in the environment, and accumulate in the bacterial membrane affecting its physical-chemical properties, and in consequence, their functions (Weber & De Bont, 1996). This could suggest a better tolerance to organic compounds, such as ethanol, for L. kunkeei MP2, which could help this organism to tolerate unfavorable conditions, and have a unique competitive advantage compared to other Lactobacillus species (De Guchte van et al., 2002).

The diversity of Firmicutes species in A. mellifera could imply a metabolic diversity that could be crucial for honey bee fitness (Engel & Moran, 2013). Comparative genomics of Lactobacillus genomes, have shown that close to 45% of its accessory genome encode for proteins involved in carbohydrate metabolism and transport functions (Ellegaard et al., 2015). With this metabolic diversity found in the accessory genome, is no surprising to find unique genes in the accessory genome of L. kunkeei MP2, when compared to other strains of L. kunkeei, as well as other Lactobacillus species (Table S3). These genes encode for proteins that take part of the degradation of carbohydrates, transport of molecules, transcription, as well as membrane proteins. It is very likely that some of these genes were acquired via horizontal gene transfer from a diverse group of organisms, including those that inhabit the gut of A. mellifera. This has been observed in the adaptation of strains of Gilliamela apicola and Snodgrassella alvi to the guts of the honey bee and the bumble bee (Kwong et al., 2014), as well as in other mammalian guts (Shterzer & Mizrahi, 2015).

Integrative and conjugative elements in MP2

Multiple mobile genetic elements (MGEs), were identified in the genome of L. kunkeei MP2, including prophages, transposons, and integrases. Several of these genes were unique to the MP2 genome, compared to the other draft genomes of L. kunkeei and other Lactobacillus strains. To explore a possible association between MGEs and the unique genes found in the genome of L. kunkeei MP2, we performed a prediction of genomic islands using Island Viewer 3 (Dhillon et al., 2015). With this approach, we found that most of the unique genes are found outside genomic islands (Fig. 1, Table S3). This could suggest either events of gene loss or ancestral transfer events in the genome of L. kunkeei MP2 (Tamarit et al., 2015).

Most of the MGEs found in the genome, had similarities to integrative and conjugative elements (ICEs), which are characterized by their prophage-like mode of maintenance (Burrus et al., 2002). To contrast this result, the uniquely identified genes in the genome of L. kunkeei MP2 were compared against the ICEberg database (Bi et al., 2012) (Table S3). ICEs commonly encode for genes that provide an increased fitness to the host, such as antibiotic resistance genes, phage resistance, and heavy metal transport (Burrus et al., 2002). In the case of MP2 we found genes that have similarities to transmembrane proteins, phage-related proteins, and antibiotic resistance mechanisms, suggesting that the incorporation and stability of these unique genes in the genome of L. kunkeei MP2, is providing an increase in the fitness of this bacterial strain in the gut of A. mellifera. Among the predicted phage-like sequences, we found one coding for a mef(A)/msr(D) resistance protein, with similarity to a sequence from Streptococcus pyogenes, involved in the resistance to macrolides (Iannelli et al., 2014). In the European Union, the usage of antibiotics, and antibiotic-containing compounds, is not permitted. However, macrolides (such as tylosin and streptomycin), are still used as a preventive treatment against Paenibacillus larvae, the causal agent of American foulbrood, in many countries (Reynaldi et al., 2010; Gaudin, Hedou & Verdon, 2012). Thus, if bees were exposed to antibiotics in their diet, it is possible that the gut microbiota may have acquired the necessary molecular mechanisms to adapt and survive in an exposed environment (Tarapoulouzi et al., 2013). This acquisition can be explained by horizontal gene transfer events from the surrounding natural environment (such as soil). Here we can find multiple bacterial and fungal species that commonly produce antimicrobial compounds, and could act as a source of these resistance genes (Alippi, León & López, 2014).

Most of ICEs coding genes are usually present within genomic islands in the host genome (Hacker & Carniel, 2001; Boyd, Almagro-Moreno & Parent, 2009), but in the case of L. kunkeei MP2, none of the predicted ICEs genes were found in the context of genomic islands according to the predictions performed with IslandViewer 3 (Dhillon et al., 2015). This could suggest the presence of previously uncharacterized ICEs, or also our current limitation in the detection of ICEs from Lactobacillus species.

Prediction of horizontal gene transfer events

To predict horizontally transferred genes we used Darkhorse (Podell & Gaasterland, 2007) to analyze the complete genome of L. kunkeei MP2. We did not consider hits to organisms within the same Phylum, to avoid false predictions, although this could lead to ignore real transfer events between more closely related organisms. A total of 19 genes were predicted to have been acquired via horizontal gene transfer (Table S3), with a normalized LPI score cutoff of 0.546. Seven of these genes had matches with the genome of A. mellifera, which a detailed look suggested as a contamination of the genome of A. mellifera with sequences from Lactobacillus species. This has been previously reported for other genome projects (Merchant, Wood & Salzberg, 2014). Only one of the genes predicted to be acquired via HGT was unique to L. kunkeei MP2 when compared to other L. kunkeei strains and other Lactobacillus genomes, which codifies for a hypothetical protein, with a best hit as a phage protein from Halomonas sp. HAL1. None of the predicted genes was found associated with an ICE or a genomic island. Although the apparent lack of genes of acquired via HGT could be explained by the genome reduction that has been observed in L. kunkeei strains (Tamarit et al., 2015), an alternative explanation is the lack of reference genomes of isolates obtained from the gut of A. mellifera. This could limit our ability to predict events of horizontal gene transfer between members of the honey gut microbiota.

Conclusions

Using long-read sequence technologies we were able to assemble the first closed genome for a L. kunkeei strain, resolving some of the assemble issues of a previous draft of this strain (Olmos et al., 2014). The comparison of the genome sequence against other Lactobacilus species, showed a percentage of genes that are unique to the MP2 strain, including metabolic key enzymes that could play an important role in the honey bee nutrition and fitness. The genome of L. kunkeei MP2 also has genes encoding for proteins involved in important roles such as adhesion, biofilm synthesis, and stress tolerance, which in addition to the presence of antibiotic resistance related genes, highlights the versatility of this bacteria to adapt to different environments, such as flowers or insect guts.

One of the features highlighted in this study is the abundance of prophages in the L. kunkeei genome. The presence of prophages in Lactobacillus is common, but MP2 has sequences unique to this strain. This is the case of a large genomic region (located in the 31,034-75,092 region), with genes encoding for several phage-related proteins, including structural and replicative components. The presence of prophages could be associated with lateral transference events, and therefore, with the acquisition of genes related with bacterial fitness. Given the high percentage of hypothetical proteins encoded in this region, a future goal for research, is the elucidation of the role for these proteins in L. kunkeei MP2.

Supplemental Information

Table S1 Complete EggNOG annotation of L. kunkeei MP2

Detailed information for each predicted CDS of L. kunkeei MP2, against the EggNOG database.

Click here for additional data file.

Table S2 Summary of the results for all the tools used in this study

This table contains the information of the prediction of all the tools used in the study, for each predicted CDS of L. kunkeei MP2. The information includes the unique genes of L. kunkeei MP2 against other L. kunkeei strains, as well as other Lactobacillus species. Also the results from the Darkhorse, IslandViewer and PHAST analysis.

Click here for additional data file.

Table S3 Blast results of PHAST predicted genes

Results of the BLAST search against NR of the genes found in one of the PHAST predicted islands.

Click here for additional data file.

Additional Information and Declarations

Competing Interests

Author Contributions

DNA Deposition

Data Availability

The authors declare there are no competing interests.

Freddy Asenjo, Juan A. Ugalde and Annette N. Trombert conceived and designed the experiments, performed the experiments, analyzed the data, contributed reagents/materials/analysis tools, wrote the paper, prepared figures and/or tables, reviewed drafts of the paper.

Alejandro Olmos conceived and designed the experiments, performed the experiments, analyzed the data, contributed reagents/materials/analysis tools.

Patricia Henríquez-Piskulich conceived and designed the experiments, performed the experiments, contributed reagents/materials/analysis tools, reviewed drafts of the paper.

Victor Polanco and Patricia Aldea conceived and designed the experiments, contributed reagents/materials/analysis tools, reviewed drafts of the paper.

The following information was supplied regarding the deposition of DNA sequences:

GenBank accession number CP012920.

The following information was supplied regarding data availability:

Figshare: http://figshare.com/articles/Lactobacillus_kunkeei_MP2_Genome/1576406.

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
