# Peer review of "Genome sequencing and analysis of the first complete genome of Lactobacillus kunkeei strain MP2, an Apis mellifera gut isolate"

_PeerJ, doi:10.7717/peerj.1950_

## Round 0.1 · original submission · Major Revisions

I consider your work a well-written manuscript with interesting information where you present the first complete genome of a strain of Lactobacillus kunkeei, a common inhabitant of the gut of honey bees. However, there is some concern rose by one of the reviewers stating that your manuscript is an improved version of a previous manuscript from 2014 (Olmos et al., 2014 in Genome announcement). This can make the novelty of your manuscript much reduced. So, I encourage you to clearly state the differences between both works and try to enhance the novelty of your research satisfying or answering some concerns raised by reviewer 2

Reviewer 1 ·

Basic reporting

A well-written manuscript where the authors present the first complete genome of a strain of Lactobacillus kunkeei, a common inhabitant of the gut of honey bees. They annotated the genome and compared the sequence with other strains of the same species and with the genome of other species of the genera Lactobacillus. This allows them to identify unique zones some corresponding to prophage genes and other to unique genes related with metabolic key enzymes and with genes encoding for proteins involved in important roles such as adhesion, biofilm synthesis and stress tolerance.

Experimental design

The procedures and tools used to conduct the work are appropriate and well used to get the expected results, though extended or additional analysis will help to better understand the content of the manuscript. In this way, alignment of genomes using MAUVE which offers a graphically environment for better identify the presence of large-scale evolutionary events such as rearrangement and inversion would be appropriate. I also miss a phylogenetic tree, for example based on the 16S rDNA, incorporating the strain MP2 and other strains of the same species and from other related species like L. rhamnosus, L. plantarum, L. brevis, etc.

Validity of the findings

Results obtained in the present work are appropiate though in order to make this manuscript more valuable, the incorporation of the information about the new draft genomes of L. kunkeei (nowadays there are at least 10 more genomes) would help to elucidate and confirm more accurately the presence of unique genes in the strains MP2.

Additional comments

L123-125, this paragraph should be re-written to make more evident that this part has been performed in a previous study. One option could be: “ L. kunkeei strain MP2 used in this study was isolated in a previous work from the gut of a honey bee (Apis mellifera) collected from a commune hive located in the Maria Pinto area at the Melipilla Province in the Central zone of Chile (Olmos et al. 2014)”.
L143-144, introduce the code of the strain used in the comparison related with the bibliographic reference.
Tables. In my opinion SUPPLEMENTARYTABLE1.XLSX containing the genome sequences used in this study should be incorporated in the manuscript.

Reviewer 2 ·

Basic reporting

The manuscript entitled “Genome sequencing and analysis of the first complete genome of Lactobacillus kunkeei strain MP2, an Apis melífera gut isolate”, by Asenjo et al., fits into scope of the PeerJ Journal. It describes sequencing and “in silico” analysis of some traits present in that bacterial genome. The English is clear, but with some typos (see specific comments).
The manuscript is classically structured, presenting a brief introduction and background, material and methods, results and discussion, conclusions and references. It is an improved version of a previous manuscript from 2014 (Olmos et al., 2014 in Genome announcement), so the novelty is much reduced and consist in that now the sequence is complete. However, conclusions are not sound enough, consisting in just few speculations based in the presence of determined sequences without any other experimental demonstration.
Figure 1 and Table 1 are relevant, but more results would be desirable. This is just general information about the genome, but other relevant information, such as comparison with other genomes, will be necessary.
In general, is a very descriptive paper, with not too many experimentation backgrounds.

Specific comments:
Pg. 4, line 56: Please, add a dot after “L”.
Pg. 7, line 127: underscript “2” of “CO2”.
Pg. 14, line 279. “GftC”, gene or protein? You referred as a gen, so it would be lowercase and in italics. The same for page 15, line 301. “YfkJ” which should be not in italics.
Pg. 17, lines 351-359. This statement needs a reference.
Reference section: scientific and gene names are not in italics.

Experimental design

Methodology appropriate but can not be reproducible, since the genome sequence is not available. Sequence should be deposited in order to confirm those findings.
Low novelty was found for this manuscript. As mentioned previously, this is just an improved version of a previous manuscript from 2014 (Olmos et al., 2014 in Genome announcement). No relevant conclusions were build up from this manuscript. I miss comparison of strain MP2 with those published by Tamarit et al., 2015 This would add more information about the MP2 strain.
The manuscript accomplish the ethic standards of the field.

Validity of the findings

Reviewers do not have access to genome data, so no way to confirm the findings.
Conclusions are just a recap of the results from the genome with speculations about the functions of group of genes found in the strain MP2. Part of these results have been published previously by Olmos et al., 2014.
The main finding of the manuscript is the obtention of a complete genome from MP2, solving few aspects that were not possible to conclude from the draft genome, but the main aspects were already published and comment in previous works.

Additional comments

I consider that this is a good beginning for a interesting manuscript. However, the unique improvement from previous works is just having the complete genome without further relevant information.
To my opinion, this manuscript would need further experimentation to contain new and further findings in order to be relevant.

---

## Round 0.2 · accepted · Accept

The authors have made a satisfactory revision of the manuscript incorporating and answering to all comments made by the reviewers.

Reviewer 1 ·

Basic reporting

As I commented in the previous review, this manuscript is well written and introduces the first complete genome of a strain of Lactobacillus kunkeei, a common inhabitant of the gut of honey bees. After the major modifications, the results are more robust and conclusions more contundent. In addition, introduction of new material allows the manuscript to be up to date with new analysis methodologies and the addition of new reference sequences.

Experimental design

In relation to the methodology used, authors successfully justify the use of alternative analysis instead of MAUVE and incorporate the suggestions proposed in the previous review, concretely, the phylogenetic tree based on the 16S rDNA.

Validity of the findings

According to previous suggestions, authors have incorporated additional 15 strains of L. kunkeei in the analysis.

Additional comments

All the proposed changes have been incorporated in the manuscript and I think that the resulting manuscript fits with the acceptable publication.